# Quantifying the Economic Impact of Bovine Tuberculosis on Livestock Farms in South-Western Spain

**DOI:** 10.3390/ani10122433

**Published:** 2020-12-18

**Authors:** Rosario Pérez-Morote, Carolina Pontones-Rosa, Christian Gortázar-Schmidt, Álvaro Ignacio Muñoz-Cardona

**Affiliations:** 1Research Group GISEIO “Sistemas de Información Externa e Interna de las Organizaciones: Información Corporativa y para la Gestión”, Business Administration and Management Department, University of Castilla-La Mancha, 02071 Albacete, Spain; carolina.pontones@uclm.es; 2SaBio Instituto de Investigación en Recursos Cinegéticos IREC (UCLM & CSIC), 13005 Ciudad Real, Spain; christian.gortazar@uclm.es; 3Spanish Association of Bovine Meat Producers in Castilla-La Mancha, Calle Corpus Christi 12, 45005 Toledo, Spain; gerenteclm@asoprovac.com

**Keywords:** animal tuberculosis, economic impact, cost of production of cattle, losses by slaughter, losses by replace, loss of profit

## Abstract

**Simple Summary:**

Spain is the fifth largest beef cattle producer in the European Union. Animal tuberculosis (TB) is one of the primary limitations of beef cattle farming in the affected countries. The closed herd policy is an effective tool against animal TB, being based on rearing the calves born on the farm for replacement. When the disease appears on a farm, a series of decisions must be taken that will cause substantial losses for three main reasons: First, the final balance generated by the slaughter of infected animals; second, the lower sales revenue derived from the replacement of the slaughtered cattle; and third, the costs incurred from immobilizing the rest of the animals in the herd. Quantifying the impact of TB is determined by the loss of profit, with the consequences being different in relative terms depending on each farm, where those with larger numbers of head of cattle are more resilient to a given number of positive cases. It also contributes to the debate about the optimal balance of cost sharing between the government and farmers.

**Abstract:**

Pasture-based livestock farming generates income in regions with limited resources and is key to biodiversity conservation. However, costs derived from fighting disease can make the difference between profit and loss, eventually compromising farm survival. Animal TB (TB), a chronic infection of cattle and other domestic and wild hosts, is one of the primary limitations of beef cattle farming in some parts of Europe. When an animal tests positive for TB, a loss of profit is caused in the farm, which is due mainly to the animal’s slaughter, replacement of the slaughtered animal and the need to immobilize the rest of the herd. We estimated the economic impact in terms of loss of profit as a result of incremental costs and forgone incomes. We show that farms with a larger number of heads are more capable of dealing with the loss of profit caused by the disease. The quantification of the loss of profit contributes to the ongoing debate on the co-sharing of TB costs between government and farmers. The compensation farmers receive from the public administration to mitigate the economic effects of the disease control interventions is only intended to balance the loss due to slaughter of the infected cattle, being the loss of profit a more global concept.

## 1. Introduction

Extensive livestock grazing is one of the remaining productive activities in the less-favored rural regions of Europe. Pasture-based livestock farming systems not only help settle rural population and generate income in fundamentally rural regions with limited resources but are also key to biodiversity conservation [1]. Spain is the fifth largest beef cattle producer in the European Union [2]. The number of beef cattle herds, mainly belonging to family enterprises, fell 17% from 2017 to 2019 [3], and farmers rely on subsidies to maintain profitability [4]. Given this scenario, incurring additional costs such as those derived from fighting disease can make the difference between profit and loss.

Animal TB (TB), a chronic infection of cattle and other domestic and wild hosts, is one of the primary limitations of beef cattle farming in affected countries. It is caused by infection with *Mycobacterium bovis* and other closely related members of the *Mycobacterium tuberculosis* complex (MTC) [5]. In cattle, the primary bronchopneumonic infection may remain localized or progress slowly for considerable periods of time, eventually leading to generalized lesions. Therefore, tuberculin positive cattle are regarded as “open” cases of TB, potentially capable of transmitting infection to other animals and humans [6]. Vaccination, or treatment of infected cattle, is prohibited by EU regulations. Thus, test-positive cattle are compulsorily slaughtered and subjected to post-mortem examination. However, in countries where there is transmission of infection from endemically infected non-cattle animal populations, eradication is compromised and control measures must be applied indefinitely [6]. Such external factors can have a significant influence on the farmers’ perception of the effectiveness of control efforts against cattle TB [7].

In TB outbreaks in Spain, the most common number of skin-test reactor cattle per herd is 1–2, with a range of up to 39 [8]. However, if MTC infection is confirmed by culture, re-testing the herd with the more sensitive gamma interferon test will yield higher numbers of TB-positive cattle [9]. In addition, the average prevalence of bovine TB in herds in both the EU (0.9% in 2018; [10,11] and Spain (2.3% in 2018) has remained stable or increased slightly in recent years, which indicates the profitability of farms can be seriously affected by the presence of repeated outbreaks of TB. Regarding marginal lands of limited productivity, a study conducted in Australia, a country that has successfully eradicated TB, found that “unless radically modified, the campaign will eradicate the sub-marginal cattlemen as well as bovine TB” [12].

Unless a national strategy offers incentives, profit-maximizing producers cannot be expected to eradicate the diseases from their herds due to the increasing marginal cost of controlling the infection at low levels of prevalence [12]. In the EU, the Commission implements a series of co-funded national programs for the eradication of TB, based on testing plans and compensation for removal of stock, control of movement and inspection of slaughtered animals [13]. However, the optimal balance of cost sharing between the State and the private sector remains a subject of debate, as in other countries [7].

Estimation of the economic impact of TB outbreaks on farmers is the main aim of this paper. This estimation can provide valuable economic information for making a decision as to whether to use control measures and assessment of their benefits. Previous studies have investigated the economic impacts of a bovine TB breakdown by estimating the total financial loss for cattle farmers caused by TB (e.g., [14,15,16,17,18] for the UK or [19,20] for the US). The approach of the respective authors differs regarding the items included in the resulting total financial loss and the different criteria, parameters and calculations used. These authors also looked at how these losses relate to incentives and reflected on the challenge of deciding the balance of cost sharing between the state and the private sector [14].

Given this scenario, our paper adds to this body of knowledge by providing a methodology and example for the estimation of the economic impacts from a TB-endemic region of Europe.

When the disease appears on a farm, a series of decisions must be taken that will cause a loss of profit. According to Spanish Law [21], the reactors must be slaughtered, and the rest of the herd can only be sold to an uncertified feedlot, with a subsequent decrease in its sale price, or fattened in the own farm, hence causing additional costs. Moreover, the replacement of the slaughtered animals by purchasing new cattle is subject to time restrictions imposed by the above legislation and is not advisable under Spanish TB control schemes [22] (PATUBES in its Spanish acronym). This means replacements should be made with calves born within the herd, with these no longer being available for sale. In this paper, we identify three components of the loss of profit due to TB: First, the final balance generated by the slaughter of infected animals; second, the income forgone and incremental costs from retaining cows within the herd and rearing cattle which would otherwise have been sold, as they are needed to replace the slaughtered animals; and third, the costs incurred from immobilizing the rest of the animals in the herd. To determine these amounts, our starting point is the cost of production of the animals, depending on age and conditioned by weight. We expect the analysis to show that farms with a larger number of head are more capable of dealing with the loss of profit caused by the disease.

The article is structured as follows: After this introduction, the second section sets out the background to the research and the theoretical model related to the cost of production and the loss of profit on livestock farms. The third section explains the materials, sources and methodologies used to obtain our results, which are presented in the fourth section. The fifth section comprises the discussion, limitations of the study and future lines of research. Finally, we present our conclusions and the bibliography.

## 2. Background and Theoretical Model

This section explains the business model of the farms under study and the actions and consequences derived from the detection of test-positive animals. We also establish the types of losses produced, using the legal concept of loss of profit.

### 2.1. Business Model

This study focuses on cattle breeding farms, defined as those that have female animals, destined to producing calves to be sold at weaning, or for fattening [23]. Particularly, the farms under study implement a closed herd policy, which means that the present study is conducted under the premise that the animals slaughtered due to TB are replaced by calves born at the same holding. There are various reasons for this strategy. On the one hand, Royal Decree 2611/1996 [21] sets out restrictions on movement in cases of TB. Specifically, following a positive test, the farm is forced to wait 6 months before conducting another test, and only if the herd tests negative can the farm purchase animals from other livestock holdings. On the other hand, PATUBES recommends that new animals entering a farm be kept to the minimum to reduce the risk of TB and warns of the risk of importing animals even if these are acquired in countries with an organized trading facility (OTF) system. The Spanish Association of Beef Producers (ASOPROVAC, Toledo, Spain) [24] argues that, apart from health reasons, farmers have a preference for replacing lost animals with livestock of their own. Breeding their own cattle means they have access to the genomic information of the animals on the farm, enabling them to select the mothers with the best characteristics in terms of fertility, handling, ease of calving, or maternal qualities. Moreover, rearing their own animals means they are able to better adapt to market needs and customer demands. The cows born on the farm are also perfectly suited to the environment in which the farm is located, and an emotional link is created between animals and breeder. Another important advantage is that this system enables the farmer to safeguard sanitary control. Breeding animals are obliged to be duly vaccinated and receive any compulsory heath treatments before their first mating [25].

The closed herd policy implies that to cover the natural requirements of replacement that will maintain the farms’ reproductive capacity, a certain number of female calves must be destined to replacement each year.

### 2.2. Actions and Consequences Derived from Detection of TB

When an animal from the herd tests positive for the disease or becomes infected, slaughter is compulsory. This means that, depending on the animal’s age, it will have been depreciated to a greater or lesser degree, according to the number of calves it has produced until that date. This further implies that, when slaughtered, part of the animal’s value will not have been recovered, thus generating a loss for the farm. This loss is, to a lesser or greater extent, mitigated or balanced by the compensation per culled animal paid by the public administration, as well as the value of the beef obtained at slaughter. 

Furthermore, detection of TB necessitates, in order to maintain the productive capacity of the farm, that an additional number of animals are used for replacement. This means that the animals due for replacement cannot be sold [26,27], entailing an opportunity cost represented by the amount of forgone income. Moreover, the estimated age of first calving is around two and a half years [28], and thus the replacement calf will only reach reproductive capacity by that time. Consequently, during this period the farm’s productive capacity will be diminished, having to bear the additional reduction of sales revenue resulting from the lower production of calves. 

In addition, test-positive animals must be isolated from the others and slaughtered as soon as possible, within a period of no more than 30 days [21]. The rest of the herd is subject to restrictions on movement, only being allowed to leave the farm for sale to an uncertified feedlot [29]. If they are not sold to this type of feedlot, or until the sale takes place, the entire herd must remain on the farm until the animals can be sold to a certified feedlot, which involves an estimated period of 18 months [24], generating additional costs from fattening and care [16,17]. The other option is to sell to an uncertified feedlot at a reduced sale price [16,30]. Uncertified feedlots are not allowed to export animals, being limited to selling in the national market. The reduced sale price to uncertified feedlots is explained by the fact that when animals need to be slaughtered in Spanish facilities, their price depends on the kilos of beef carcass, which, in turn, depends on the actual carcass yield, further discounting, transport costs, fees, and cost of slaughter [24].

Finally, the Spanish central veterinary service provides the diagnostic test at no cost, not resulting in additional expenses for farmers. 

### 2.3. Theoretical Model and Conceptual Framework

The legal concept of loss of profit introduced by the Civil Code (art.1106) [31], is defined as a form of property damage that consists of the loss of a legitimate profit or economic benefit by the victim or their family members as a result of an injury (consequential damage), which would not have occurred if the harmful event had not been verified. Loss of profit occurs when there is a loss of a certain prospect of benefit. For example, the trader whose merchandise has been destroyed can claim the price of the merchandise as well as the profit they would have made. Article 1106 of the Civil Code states that “compensation for damages includes not only the value of the loss suffered, but also the value of the profit that the creditor has ceased to obtain”.

In order for compensation for loss of profit to be awarded, case law requires two conditions are met: That the lost profit exists and can be proven, along with its direct relationship to the harm caused, and that the amount that has been lost can be economically determined. Although these requirements tend to be demanded, the most recent jurisprudence of the Civil Chamber of the Spanish Supreme Court is opting to compensate “that future loss which can reasonably be foreseen to occur”. In order to carry out the “quantum” (amount) of the compensation for loss of profit, when it refers to future profits, it must be obtained through prospective assessments, based on objective criteria of experience, among which those operating in the economic, accounting, actuarial, assistance, or financial world can serve, according to the corresponding technical or scientific disciplines, in accordance with the examination and weighting of the circumstances of each case. 

Specifically, the financial accounting discipline set out in the Spanish General Accounting Plan [32] provides for criteria of assessment and regulations to ascertain loss of profit, whereas analytical accounting provides for guidance to design and control company costs [15,33,34,35]. 

Figure 1 below sets out the theoretical accounting model for the cost of production of livestock farm assets. It can be seen that a first asset (cow, Asset 1) permits the production of calves (Asset 2) destined for sale or replacement.

The theoretical concepts included in the model are the following:Cost of production, which includes the purchase price of raw materials and consumables, costs directly attributable to production of the asset and the proportional amount of production costs indirectly attributable to the asset, insofar as these were incurred during the production, construction or manufacturing period. They are based on the level of usage of normal production capacity and are required to bring the asset into operating condition. It should be noted that cost calculation is characterized by its relativity, given the need to decide between different criteria when distributing costs [33], and thus the figures reached should be understood as estimations and not unique, categorical values.Purchase price, which comprises the amount invoiced by the seller, after deduction of any discounts, rebates or other similar items, such as interest incorporated into the nominal amount, plus any additional costs incurred to bring the goods to a saleable condition, such as transport, import duties, insurance and other costs directly attributable to the acquisition of inventories. Nevertheless, the purchase price can include interest on payables maturing within one year which do not have a contractual interest rate when the effect of not discounting the cash flows is immaterial [32]. This first asset (Asset 1) is assessed on this criterion.Residual value of an asset, understood as the amount that the company would currently obtain from disposal of the asset, after deducting the costs of disposal, if the asset were already of the age and in the condition expected at the end of its useful life [32]. In the field of livestock holdings, the residual value of cattle corresponds to their cull value.

The difference between the purchase price and the cull value is the depreciable value the company will need to recover over the useful (reproductive) life of the animal, understood in this case as the period during which the cow is fertile.

As Asset 1 is maintained on the farm for the purpose of calving, its cost must be factored across the calves it produces. In this sense, the ratio between the depreciable value and useful life of this first asset represents the annual depreciation amount to be recovered in each of the years of useful life. This amount, together with the direct and indirect costs of maintaining the asset, represents the value of Asset 2 (newborn calf). From these concepts, we must subtract the amount of the official subsidies received by these farms, which translate into lower costs for the farms [35]. The resulting Asset 2 (newborn calf) may be sold or destined for natural replacement. Its initial resulting cost will increase depending on the time it remains on the farm as a consequence of the direct and indirect costs generated by its maintenance.

Figure 2 shows the theoretical accounting model for the loss of profit, comprising three components: The final balance generated by the slaughter of infected animals; the income forgone and incremental costs due to the replacement of the slaughtered animal; and third, the costs incurred from immobilizing the rest of the animals in the herd:

Regarding the first component, the final balance generated by the slaughter of positive reactors, the following accounting concepts are involved:Unrecovered value of an asset, the slaughtered cow in this case, is obtained by multiplying the outstanding useful life of the animal by its annual depreciation [32,35].Residual value or cull value, which was previously explained.Incomes, represented by the compensation received for the slaughter. Following the ICAC (2007) [32], grants to finance specific expenses shall be recognized as income in the reporting period in which the financed expenses are accrued [32].

The second component of the proposed model is twofold: Reduction in sales revenues and incremental cost of rearing replacement calf. This involves the following accounting concepts:Fair value, as a measure of sale price. *Fair value* is the amount for which an asset can be exchanged, or a liability settled, between knowledgeable, willing parties in an arm’s length transaction. Fair value shall be determined without deducting any transaction costs incurred on disposal. The amount a company would receive or pay in a forced transaction, distress sale or involuntary liquidation shall not be considered as fair value [32].Cost of production is based on the definition previously explained and will include all the direct and indirect costs generated in order for the Asset 2 (replacement calf) to be in the same conditions as the asset lost [36,37].

Regarding the third component, the costs derived from immobilizing the rest of the herd, it includes various accounting concepts: Additional costs of production and fair value, taking the existence of disease into account insofar it affects the sale price.

## 3. Materials and Methods

Our study region was western Spain, from Zamora province in the North to Cádiz in the South, which accounts for about 60% of the Spanish beef cattle population, reared in a characteristic habitat of mixed agriculture, pastureland and Mediterranean woodland [26].

### 3.1. Materials and Methods for the Estimation of Cost of Production

Following the flow diagram presented in the previous section in Figure 1, the starting point was estimating the production costs of the calves born and reared on these farms (Asset 2), following a methodology similar to that used in previous studies, such as the those by the MSD Group (2019) and Burh, McKeever and Adachi (2009) [19,38]. To this aim, we chose a cow aged between 48 and 84 months as representative of the first asset (Asset 1), given that the mean age of cattle livestock is 6.72 years [28]. Using the purchase price, cull value and useful reproductive life, we calculated the annual depreciation per cow for each year of the “cow” asset’s useful life. To this quota, we added the direct cost of feeding and health care, salaried labor, and indirect costs. We then subtracted the mean amount of direct or per head subsidies received. The total represents the annual cost of owning and maintaining a cow. This ownership and maintenance of a cow translates into calving, accounting for fertility rate, calf production and calf mortality at birth. As a result, we obtain the unit cost of production per calf at the time of calving.

The next step was to calculate the unit cost of production in each of the age categories. To do this, we took as a starting point the previously obtained unit cost of production of the newborn calf and, depending on the average number of months that each animal remains on the farm generating costs, we added the monthly amount of the costs of feeding and health care, salaried labor, and indirect costs. We then subtracted the monthly amount corresponding to direct subsidies received. It should be noted that we considered that until the age of 6 months (unweaned calf), calves generate no additional labor salaries or indirect costs with respect to those of their mother. Consequently, we allocated the cost of only feeding and health care.

We now describe the sources which served as a basis to determine the values required to estimate the cost of production referred to above. The purchase price was obtained using as reference the prices provided by the National Cattle Market in Talavera de la Reina (Spain) as of 31 December 2019 [39]. These prices are, in some cases, per physical animal (as in the case of calves under a month old), while others are given per kg of weight. For the categories whose purchase prices depends on weight, we calculated a mean price of 2.23 €/kg for yearlings to the age of 18 months. From this age on, live animal purchase prices are per animal, ranging between 1500 € for a yearling aged 18–24 months and 1050 € for a heifer. To determine the residual or cull value of the older cattle, we drew on the same source, using beef carcass meat prices, considering a carcass yield of 57.5% [40]. These values vary between 1.99 €/kg for the categories of calves, yearlings and heifers, and 0.52 €/kg for heifers and cows.

All the above data were compared and complemented with other technical data provided by ASOPROVAC (Toledo, Spain). The mean weight, which depends on sex, age, and breed, was calculated using data provided by the Spanish Ministry of Agriculture [41]. We also considered the loss of body condition in the older animals [42].

Throughout the study, and to determine the cost of production, we distinguished between five categories according to age, as follows: “Un-weaned calves”, aged up to 6 months; “weaned calves” (animals aged 6–12 months); “yearlings” (animals aged 12–24 months); “heifers” (animals aged 24–48 months); and “cows” (animals > 48 months) [17,39]. In order to conduct our analysis, some of the categories were further divided into more detailed sub-categories, in line with the provisions of Royal Decree 389/2011 of 18 March [43]. Thus, in the 0–6-month age range, which identifies unweaned calves, we differentiated between a calf ≤ 1 month, 1–3 months, 3–6 months. In the 6–12-month age range, we differentiated between animals aged 6 to 9 months and animals aged 9 to 12 months. In the 12–24-month age range, corresponding to “yearling”, we differentiated between animals aged 12 to 18 months and animals aged 18 to 24 months. In the >48-month age range, that is, the “cow” category, we discriminated between animals aged 48–84 months, 84–120 months, and >120 months.

Table 1 shows the values used for each category and age:

In addition to the above, in order to determine the production costs (direct and indirect), we collected secondary data from different official sources. The cost data used to calculate production costs were taken from a series of reports published by the Spanish Ministry of Agriculture “Studies on Costs and Income of Agricultural Holdings” [44,45,46,47,48]. This source provides data at head of cattle level on farms belonging to different autonomous communities, without the possibility to differentiate between types of holdings. We used the mean values from 2013 to 2017, adjusted to prices as of 31 December 2019. Estimation of direct cost included a mean cost of feeding the animals and also their health care and labor costs. Feed cost essentially comprises the sums for forage and concentrates corrected for the mean variation of inventories. For the labor factor, we considered the cost of salaried labor.

Regarding the indirect costs that were taken into account, they included those related to hiring machinery, depreciation of fixed assets, social security, insurance, taxes, and general costs [49,50,51].

Finally, the sum of direct and indirect costs was reduced by the mean amount of direct subsidies received from the Common Agricultural Policy.

### 3.2. Materials and Methods for the Estimation of Loss of Profit

In order to estimate the first component of the loss of profit, the final balance per slaughtered animal, we calculated the difference between the animal’s unrecovered value on the date of slaughter, and the amount received, which is made up of the cull value and the compensation. Moreover, to determine an average final balance, we estimated a weighted mean, considering that a typical herd of beef cattle generally comprises around 25% of animals aged 0–24 months and 75% aged over 24 months.

The sources from which we took the data required to estimate the final balance due to slaughter were the following:

Royal Decree 389/2011 of 18 March [43] provided us with the compensation to be received, according to age, for each test-positive animal on the farm, which was used to determine the final financial balance due to slaughter.

The cost estimations performed took into account the parameters for the useful life estimated in 11.5 years, given a life expectancy of 13–14 years [28]. With a fertility rate of 72.8%, each female calf produces 9.05 calves over their lifetime [28]. We assumed calf mortality of 5% [52] and a mean replacement percentage of 12.5% [26,27].

As regards the second component of loss of profit, it consists of two elements: The reduction in sales revenue and the incremental cost of rearing replacement calf until they become heifers. Concerning the latter, these costs account for an annual amount in feeding and health care, salaried labor and indirect costs, minus the corresponding direct subsidies. This cost is projected over two and a half years until the replacement calf is able to calve for the first time.

To determine sales revenue, we considered the number of calves produced and sold according to the previously established fertility, mortality and replacement needs, distinguishing between the male and female animals born. This proportion is practically 50% [28], although the sale price of a male (676.97 €/calf) is higher than that of a female (458.1 €/calf). To calculate this, we used quoted market prices as of 31 December 2019. It is worth noting that the rate of variation in these prices over the last ten years is 8.6% [39]. It should also be noted that these unit sale prices correspond to a zero TB scenario. Given there is no source of reference for the sale price to uncertified feedlots, we were obliged to use the market price.

The reduction in sales revenue was identified by comparing the situation in a zero TB situation with that of the existence of 1 or more test-positive animals on the farm. We performed this comparison in four possible scenarios corresponding to different farm sizes. For the first scenario, we chose a farm with 25 head of cattle, given that 22% of the farms included in our study had between 20 and 30 head of cattle [44,45,46,47,48]. Second, we chose a scenario for a 60-head farm, with this being the average size of Spanish cattle farms in the years under study [44,45,46,47,48]. We then proposed two scenarios for farms with more than the mean number of animals, where the third scenario estimates the losses for a 100-head farm, and the fourth for a 500-head livestock holding. It is also worth noting that each of these scenarios make reference to a different number of test-positive animals, with this being higher for the farms with a greater number of cattle. It should be considered that, while most outbreaks have few positive cattle, this is not the case if there is culture confirmation. In the event of culture confirmation, the more sensitive gamma interferon test is used in addition to the skin test, and when this additional test is used, it is no longer exceptional to find larger numbers of TB-positive cattle. Moreover, we also acted on the assumption that the greater the number of head of cattle on the farm, the greater would be the likelihood of a large number of test-positive animals [9].

Within each of these four scenarios, we first calculated the yearly calf sales revenue, which depends on the calves born according to the number of productive cows. This in turn is conditioned by the number of infected animals on the farm.

The methodology used, based on secondary data, does not enable the third component of loss of profit, that is, immobilization of the herd, to be estimated. This is because of the lack of a reference market to determine the sales price to uncertified feedlots. It is also not possible to provide a mean value of the costs generated by fattening undertaken on the farm, given that these vary greatly across holdings [53,54,55].

Finally, we obtained the global loss of profit, considering the two quantified components used in this research: The final balance for slaughter and the loss from replacement. This figure is calculated for the periods of a year and two and a half years, given that some of the components of loss of profit have an effect over two and a half years.

## 4. Results

This section presents the results obtained for the cost of production of a newborn calf (Asset 2), which, in turn, is used as a basis to calculate the loss of profit, following the models described in Section 2.3., Theoretical Model and Conceptual Framework. 

Estimating the cost of production (Table 2 below) takes as a starting point the first asset (Asset 1). Using the purchase price, cull value and useful reproductive life, we calculated the annual cost of owning and maintaining a cow (554.20 €/year). This translates into calving, being the unit cost of production per calf at the time of calving estimated in 583.37 €.

From this unit cost of production for a newborn calf, we then calculated the unit cost of production in each of the age categories (Table 3). Thus, by way of example, taking into account the unit cost per calf at the moment of birth, and considering maintenance costs over time, producing a heifer of 24 to 48 months of age would cost 1463.73 €, attributing maintenance costs equivalent to an average stay of up to 48 months on the farm.

Having obtained these costs, the rest of this section presents the results of the economic valuation of the loss of profit.

Table 4 shows the average final balance per slaughtered animal. It can be seen that this final balance varies according to the category, with, in some cases, the slaughter producing not a loss, but a profit, for the farmer, as occurs in the case of animals aged 6 to 9 months, 9 to 12 months, 12 to 18 months, and 18 to 24 months. In these cases, the total of the compensation and the value of the animal at slaughter is higher than the undepreciated value. Thus, we took an average final balance per slaughtered test-positive animal of 270 €.

The second component of loss of profit consists of two elements: The reduction in sales revenue and the incremental cost of rearing replacement calf until they become heifers. As regards the latter, these costs account for an annual amount of 503.06 €, which is equivalent to annual direct and indirect cost, subtracting annual direct subsidies received. It should be noted that this cost is projected over two and a half years until the replacement calf is able to calve for the first time.

As regards the reduction in sales revenue, Table 5, Table 6, Table 7 and Table 8 permit us to compare the situation in a zero TB scenario with that of the existence of 1 or more test-positive animals on the farm. Each of these tables display the four scenarios studied: 25, 60, 100, and 500-head farms. They show the number of male and female calves sold, once those used for replacement are subtracted, whether for natural reasons or due to the replacement needs arising from the TB outbreak. The total number of male and female calves destined for sale, multiplied by their respective sale prices, determine the sales revenue of each farm, which logically decreases as the number of test-positive animals on the farm rises.

From the above Table 5, Table 6, Table 7 and Table 8, it can be seen that, in the case of there being no infected animals, a total of 16 calves are sold on the 25-head farm, 37 calves are sold on the 60-head farm, 61 on the 100-head farm, and 309 on the 500-head farm. These figures decrease as the number of productive animals falls due to TB. As an example, if we focus on Table 5 and Table 6 that, in the case of 3 test-positive animals, 1 less male calf and 5 fewer female calves are sold, compared to a disease-free scenario in the cases of farms with 25 and 60 heads of cattle. In financial terms, the annual drop in sales revenue on these farms corresponds to 2967.47 € and 7418.68 € for a period of two and a half years. In Table 7, for a 100-head farm and three test-positive animals, 1 less male calf and 3 fewer female calves are sold. In financial terms, the annual fall in sales revenue is calculated to be 2051.27 € and 5128.17 € for a period of two and a half years. In Table 8, for a 500-head farm and three test-positive animals, 1 less male calf and 4 fewer female calves are sold. In financial terms, the decline in annual sales revenue is estimated at 2509.37 € and for a period of two and a half years, this drop is estimated at 6273.42 €.

To sum up, Table 9, Table 10, Table 11 and Table 12 show the economic valuation for the loss of profit, considering the two quantified components used in this research. The figures are calculated for the periods of a year and two and a half years, given the long-term impact of some components.

If we observe Table 9, Table 10, Table 11 and Table 12, the first component of the loss of profit is the final balance due to slaughter. This element only affects the financial period in which the animal tests positive and is slaughtered. Its amount was estimated in Table 4 and was on average 270.3 € for the case of only one test-positive animal detected on the farm. If the number of positive cases is higher, then this value is multiplied by the number of animals slaughtered. A second component of the loss of profit is related to replacement. On the one hand, this includes the opportunity cost in terms of the non-sale of the replacement and the income the farm loses due to the non-birth of calves during the time between the slaughter and the replacement calf calving for the first time—which are projected over two and half years, during which time there is a fall in the farm’s productive capacity. On the other hand, it contains the incremental cost of rearing replacement calves until they become heifers, which is estimated as an annual amount of 503.06 € in the case of only one test-positive animal, while for a period of two and a half years this would amount 1257.65 €.

Regarding the calculation of overall loss of profit both for a year and for two and a half years, and taking as an example the case of three test-positive animals on the farm, this amount is estimated to be 5287.61 € for a year and 12,002.58 € for two and a half years in 25- and 60-head farms (Table 9 and Table 10). For a 100-head farm (Table 11), these figures are 4371.41 € and 9712.08 € for the respective periods. In the case of a 500-head farm (Table 12) with three test-positive animals, the loss of profit is calculated to be 4829.51 € and 10,857.33 € for one year and two and a half years, respectively.

Figure 3 shows the comparison between the four proposed scenarios with a number of up to five test-positive animals:

From Figure 3, it can be seen that, in absolute terms, the economic impact of the disease is similar in the four proposed scenarios, although, in relative terms, this impact is much more dramatic in farms with a lower number of cattle. The largest farms suffer a lower economic impact as their overall revenue helps soften the impact, increasing the resilience of such farms and enhancing their likelihood of surviving outbreaks of the disease compared to smaller livestock holdings.

Finally, it is worth underlining the effect an outbreak has on farms in relation to the possibility of continuing with their closed herd policy, maintaining their productive capacity. This effect is depicted in Figure 4.

Figure 4 shows that once a certain threshold of infected animals is passed—5 positive cases in the 25-head farm, 10 positive cases in the 60-head farm, 18 in the 100-head farm and 90 in the 500-head holding—the replacement needs exceed the farm’s production of female calves. This means that to uphold it productive capacity, the farm would be obliged to acquire animals from an outside source. In general terms, and considering the four scenarios, this occurs when the number of test-positive animals in the farms exceeds a range of between 17% and 20%. It can be observed that the impact of the disease, for a low number of positive cases, has a much more damaging effect on small holdings than on larger beef farms.

## 5. Discussion

In Spain, the successive sanitation campaigns, together with the other measures included in the national program designed to control bovine TB, has reduced the prevalence in herds since the year 2000, although the aim of declaring the whole country officially TB-free is still far from being achieved. The Spanish autonomous communities with the highest incidence of bovine TB are those in the south-west, where the present study was conducted.

There are empirical difficulties associated with the control of a disease for which an effective wildlife reservoir exists [56]. In Spain, the national and regional animal health authorities are promoting actions aimed at the primary maintenance hosts of the MTC in the Iberian Peninsula, namely, the wild ungulates, including wild boar, red deer, and fallow deer [57]. In addition, some regions have implemented, or are currently implementing, TB control plans for goats. However, some wild TB hosts, such as badgers, and some domestic maintenance hosts, are essentially excluded from the intervention. In light of this situation, successful eradication of the disease remains challenging and cattle farms will have to continue to tackle occasional TB outbreaks, seriously impacting their profitability, particularly in high-prevalence regions.

Given that TB will continue to be a risk for beef cattle operations in many regions of Europe, alternatives are needed to avoid the closure of businesses. Such alternatives include promoting biosafety interventions to reduce the risk of TB, developing integrated anti-TB plans designed to control infection in the appropriate host reservoirs (wild and domestic) and so interrupt the primary chains of transmission.

Royal Decree 138/2020 [22], which resulted from the Action Plan against TB in Wildlife Species [57], provides for actions to enhance biosafety on beef farms in Spain. Although the implementation of these measures can sometimes be complex and imply investment and costs for the livestock farmer, most of them are less costly than the loss of revenue due to the loss of just one suckler cow as a consequence of TB. Our work provides an estimation of the economic impact of the TB control campaigns, the magnitude of which evidences that prevention through biosafety is key to avoid loss of profitability and safeguard farm viability.

This research also highlights a rearing model based on a closed herd policy consisting of replacing the slaughtered animals with calves born within the same farm. Despite purchasing heifers being less costly than rearing them, when TB is present in a herd, buying is not always permitted, or at least not immediately, as shown in the studies by Benedictus, Dijkhuizen and Stelwagen (1987) [58] or Bennet and Cooke (2006) [16]. Neither is purchasing new animals the option preferred by many farmers, and, in any event, it is not advisable from the perspective of controlling infection. Consequently, closed-herd models are regarded as a good formula to protect farms against outbreaks of disease.

The valuation of the economic impact of the disease has used an approach based on principles of financial and analytical accounting applied to the reality of the farms under study. A model was proposed to estimate the loss of profit, which consisted of three components: The final balance due to slaughter, the loss related to replacement and the cost of immobilizing the herd. Some of these components affect not only the year of slaughter but their impact continues across a period of two and a half years. Our approach is consistent with that proposed by Smith, Tauer, Sanderson, and Gröhn (2014) [59], who provided a model for the cost of bovine TB in a US cow–calf herd. These authors pointed out that, at farm level, the cost of an outbreak consisted of the unrecovered expenditures due to removal of calves testing positive, the cost of replacing test-positive adult animals, and the cost of quarantine.

In any event, a methodology to estimate the loss of profit represents a contribution that helps determine the optimal balance of cost sharing between the State and farmers. Although the losses derived from the slaughter of infected animals are mitigated by the compensation provided for in Royal Decree 389/2011 [43], these amounts are not in any way intended to cover the overall loss of profit generated, as they are only conceived of as a compensation for slaughtered animals. Nonetheless, the recent announcement of complementary subsidies to make up for loss of profit in the autonomous community of Galicia evidences for the first time that the administration recognizes the existence of a loss of profit that goes beyond the simple loss of slaughtered animals [60]. Such subsidies, however, are not available to all farmers in Spain. In the UK, the recent report from the Department for Environment, Food and Rural Affairs (2019) [61] states that whereas direct impacts in the form of culled animals are routinely monitored and compensated for, other costs arising as a consequence of a breakdown are not. This is in line with our proposal, intended to highlight the existence of losses beyond those derived directly from the slaughter of infected animals, which have been widely recognized in the previous literature. Nonetheless, it should be noted that our work implements a calculation of the loss of profit that under no circumstance should be considered exact nor be evaluated in absolute terms. The wide variety of farms, the different sizes of outbreaks and the associated costs underline the difficulty of quantifying the economic impact.

Apart from the final balance due to slaughter, in farms that apply a closed herd model, the loss of profit is mainly due to the replacement of the slaughtered heifers and the recovery of the productive capacity of the farm. This generates both losses of possible revenue and incremental costs. Previous works, such as that by Hasonova (2006) [55] for paratuberculosis (Johne’s disease), identified a decrease in income because the production of the lost animal could not be replaced immediately. Moreover, our results show that, once a certain threshold of infected animals is passed -between 17% and 20%, the replacement needs exceed the farm’s production of female calves. If this situation is reached, the rearing model is no longer viable, and it is necessary to purchase new animals from outside the farm to restore the holding’s productive capacity. It should also be noted that depopulation is never the best option for the farm owner due to the reduction in the production of calves, and therefore income, for an extended period of time. However, it is in the best interest of the government to depopulate infected farms in most regions. The difference between farm and government preferences requires consideration and this potential conflict should be accounted for by disease control policy [59].

In addition, the detection of positive cases entails selling the calves to an uncertified feedlot, with the sale price of these animals being lower than the normal market price, or alternatively, the famer having to assume the costs of setting up an uncertified fattening unit on their own farm. In this regard, it has been suggested that, over time, in the case of infections with low within-farm prevalence and a slow rate of disease spread, such as TB, the cost of any movement ban is likely to exceed the epidemiological benefits, and thus, movement controls should be carefully shaped to the epidemiological and economic consequences of the disease [62,63].

The type of methodology used, based on official secondary sources, together with the great variability of the types of costs derived from immobilization, advises against quantifying this component of loss of profit, thus constituting one of the limitations of our work. Other works have managed to provide estimations of this concept, as is the case of Bennet and Cooke (2006) [16], who accomplished an on-farm survey of 151 cattle farmers that had experienced a TB outbreak and identified costs derived from isolation and movement restrictions. The isolation of reactors and inconclusive reactors involved additional labor requirements, additional straw for bedding them, costs incurred for the valuation and removal of the cattle, for example, in loading and administration. Their results highlighted that these costs would depend on the duration of the period of isolation and be conditioned by where the cattle were isolated. As regards the effects of the movement restrictions, these authors point out that extra livestock had to be retained, the numbers ranging from a few extra calves or cull cows to an entire beef enterprise. The animals could have been sold under special license, a procedure which involves some administrative and paperwork costs. Farmers reported costs due to marketing problems, restricted outlets, and the devaluation of cattle. Only 13.5 per cent of the beef farmers reported any quantifiable costs associated with movement restrictions. Other authors, such as Temple and Tuer (2000) [64], examined the consequential losses of a breakdown associated with the movement restrictions. Additionally, Butler, Lobley, and Winter (2010) [17] undertook eight case studies in England and estimated the impact of keeping stock accrued in extra costs such as additional bedding, feed and labor required to keep stock on the farm. The relative length of time that these farms had to retain their stock was also influential. Other costs were much more difficult to determine, particularly the costs of either under-utilizing or over-grazing pastures for farms with more than one holding or land that is not adjacent to the main farm. In one of the cases, a small rented farm, these additional costs became binding since margins were extremely tight and therefore cash flow became problematic.

Our results reveal that the economic impact rises depending on the number of test-positive animals, which is consistent with Bennet and Cooke (2006) [16]. For smaller farms, the loss of profit causes relatively greater damage. It was found that the larger the herd size, the more resilient was the farm, with their greater volumes of economic activity allowing them to better assimilate and mitigate the fall in profitability. Nevertheless, the economic impact cannot be considered in isolation, given that a number of studies have identified herd size as a major risk for bovine TB (e.g., [9,20]). Large herds tend to graze across larger areas, where it is easier to come into contact with infectious wildlife or livestock. Farmers may purchase and move more cattle, which facilitates cattle–cattle spread.

It should be noted that the data sources [44,45,46,47,48] used in the present work have some limitations, given that the figures for costs and others amounts, such as the subsidies awarded, were not available for each farm. This forced us to use mean data from different areas of south-western Spain referring to the period 2013–2017, with not more recent data having been published. Moreover, despite being aware that sale prices for infected herds are lower than the market price in normal conditions, it was not possible to identify such prices. This further means that the amount of lost sales revenue would be higher than the figures calculated.

Future research lines could overcome these limitations, addressing the estimation of loss of profit using primary data that facilitate greater understanding of the actual effects that detecting TB cases has on livestock farms.

## 6. Conclusions

When an animal tests positive for TB, a loss of profit is caused in the farm, which is due mainly to the animal’s slaughter, replacement of the slaughtered animal and the need to immobilize the rest of the herd.

As regards the animal’s slaughter, the loss of profit is related to the loss incurred due to the slaughter itself, which is to a lesser-or-greater extent mitigated by the compensations received. Replacing the animal means assuming an opportunity cost in terms of the income forgone by having to use healthy animals for replacement, rather than selling them. Meanwhile, the number of calves born and sold will not be recovered until the female replacement calf reaches its reproductive years, meaning a fall in sales revenue over this 30-month period. Furthermore, rearing the replacement animal generates additional costs that continue until the age of two and half years, at which point the heifer is expected to calve for the first time, with the farm thus recovering its reproductive capacity from before the disease was detected.

The larger the farm, the greater is its capacity to endure an increase in the number of infected cattle. When the number of diseased animals accounts for around 17%–20% of the herd, the female calves that are born are insufficient to cover the needs of natural replacement and of that caused by TB. Although the absolute value of the economic impact does not vary greatly according to farm size, the relative effect of this impact is much more negative for smaller farms. In larger farms, the income from sales makes it easier to absorb this impact, enhancing the farm’s resilience.

Quantifying the impact of TB on beef cattle farms may be considered a useful tool to help farmers make decisions on the advisability of investing in, and using, the measures of biosafety and biosecurity available to curb the disease on their farms. It would also be a useful tool for public administrations to help them calculate the compensation to be awarded to farmers to mitigate their losses.

## Figures and Tables

**Figure 1 animals-10-02433-f001:**
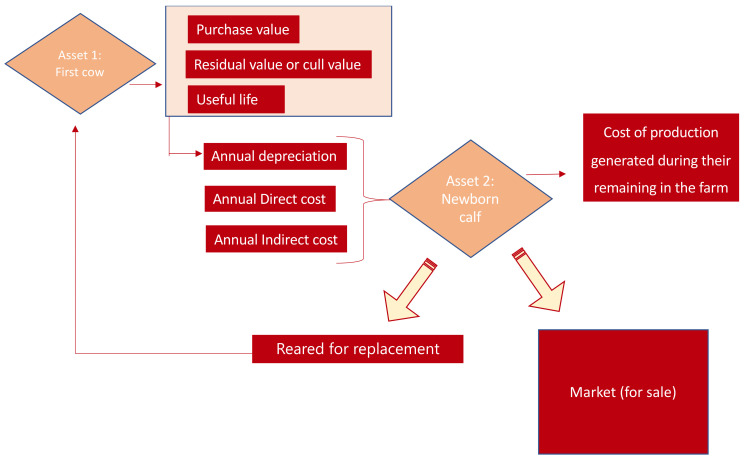
Model for estimation of cost of production.

**Figure 2 animals-10-02433-f002:**
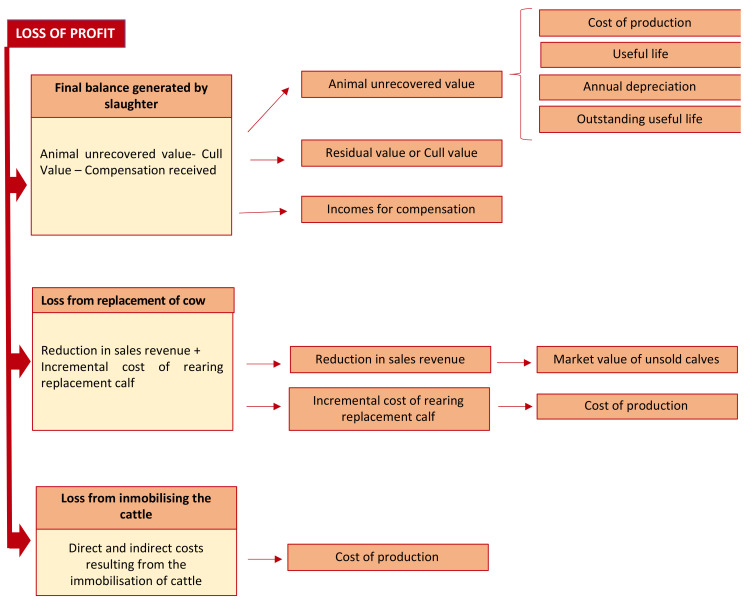
Model for estimation of the loss of profit.

**Figure 3 animals-10-02433-f003:**
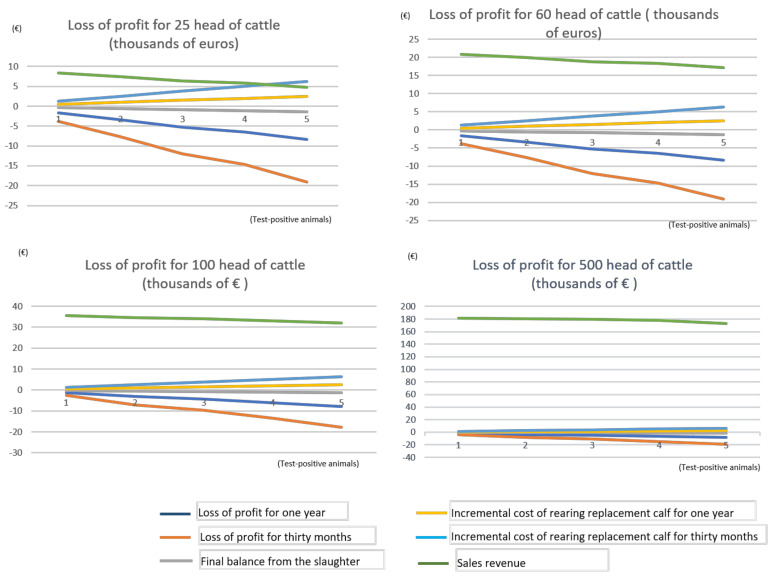
Comparative loss of profit in 25, 60, 100, and 500 head of cattle.

**Figure 4 animals-10-02433-f004:**
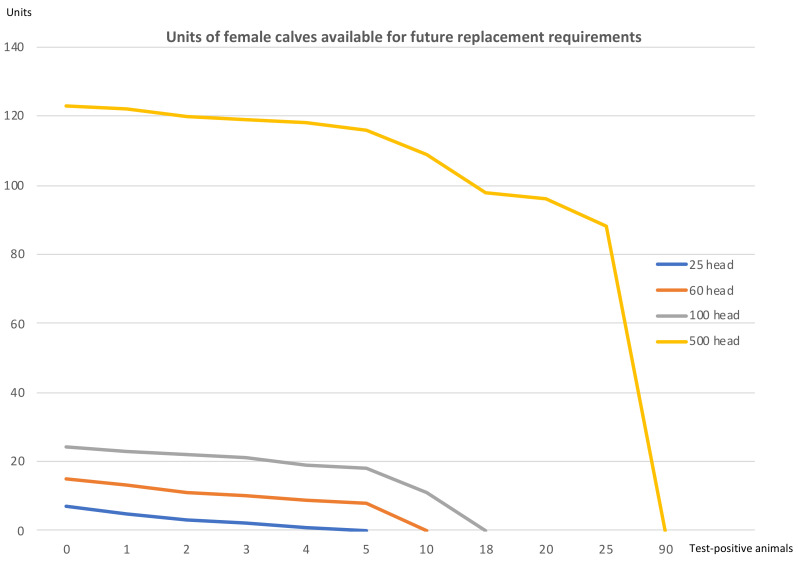
Units of female calves available for future replacement requirements in 25, 60, 100 and 500 head of cattle.

**Table 1 animals-10-02433-t001:** Category, age, weight, live purchase price and carcass value. Source: Own preparation based on data from the Talavera de la Reina Market as of 31 December 2019.

Category	Age		Live Weight (kg)	Live Cattle Purchase Price (€/unit) or (€/kg)	Carcass Sale Price (€/kg)	Total Purchase Price (€)	Cull Value * (€)
Un-weaned calves	0 to 6 months	1 month or younger	53.64	163.75 (€/unit)	1.99	163.75	61.37
		1 to 3 months	116	2.23 (€/kg)	1.99	258.68	132.73
		3 to 6 months	208.5	2.23 (€/kg)	1.99	464.96	238.58
Weaned calves	6 to 12 months	6 to 9 months	301.5	2.23 (€/kg)	1.99	671.23	344.42
		9 to 12 months	394	2.23 (€/kg)	1.99	878.62	450.83
Yearling	12 to 24 months	12 to 18 months	532.8	2.23 (€/kg)	1.99	1188.14	609.66
		18 to 24 months	632.5	1500 (€/unit)	1.99	1500.00	723.74
Heifer	24 to 48 months	24 to 48 months	640	1100 (€/unit)	1.99	1100.00	732.32
Cow	>48 months	48 to 84	640	1050 (€/unit)	1.26	1050.00	461.84
		84 to 120	640	1050 (€/unit)	0.52	1050.00	191.36
		>120 months	526.15	1050 (€/unit)	0.52	1050.00	157.32

* Cull value = 57.5% of Live weight × Carcass sale price.

**Table 2 animals-10-02433-t002:** Cost of production for a newborn calf.

Newborn Calf Production Cost		
		First Cow
Useful life	Years during which the cow is fertile	11.5 years
Purchase value		1050.00 €
Cull value		461.84 €
Annual direct costs	Feeding and health care	438.00 €
	Salaried labor	62.84 €
Annual indirect costs	Depreciation of assets, social charges, insurances, taxes and general expenses	161.98 €
Annual direct subsidies received	Subsidies received	159.76 €
Depreciable value	(Purchase price − Cull price)	588.16 €
Annual depreciation per cow	Depreciable value /useful life	51.14 €
Annual cost per cow	Depreciable value + direct cost+ indirect cost − Subsidies	554.20 €
Activity of cow across useful life (calf production)		9.00 calves
Of these 9 calves	5% Mortality	0.050
	12.5% Replacement of yearlings	0.125
	82.5% To be sold	0.825
Unit cost of production newborn calf	Annual cost per cow/0.95	583.37 €

**Table 3 animals-10-02433-t003:** Production cost of animals for reproduction by age.

Age	Calf Cost of Production (€) *
1 month or younger	619.87
1 to 3 months	619.87
3 to 6 months	619.87
6 to 9 months	638.40
9 to 12 months	709.14
12 to 18 months	834.90
18 to 24 months	960.67
24 to 48 months	1463.73
48 to 84 months	2218.32
84 to 120 months	2972.91
>120 months	3979.03

* Calf production cost = Unit cost of production newborn calf + (average number of months on the farm × (monthly direct and indirect cost − subsidies received)).

**Table 4 animals-10-02433-t004:** Unrecovered value, cull value, compensation received, final balance due to slaughter for each age category and mean final balance.

Slaughtered Animal	Cost of Production (€)	Cost of Production per Year of Useful Life * (€/year)	Outstanding Useful Life at Slaughter (years)	Animal Unrecovered Value (€) **	Cull Value (€)	Compensation Received (€)	Final Balance by Slaughter *** (€)
1 month or younger	619.87	53.90	11.5	619.87	61.37	125.06	−433.44
1 to 3 months	619.87	53.90	11.5	619.87	132.73	177.65	−309.49
3 to 6 months	619.87	53.90	11.5	619.87	238.58	341.64	−39.66
6 to 9 months	638.40	55.51	11.5	638.40	344.42	341.64	47.66
9 to 12 months	709.14	61.66	11.5	709.14	450.83	409.50	151.20
12 to 18 months	834.90	72.60	11.5	834.90	609.66	409.50	184.26
18 to 24 months	960.67	83.54	11.5	960.67	723.74	437.30	200.37
24 to 48 months	1463.73	127.28	10.5	1336.45	732.32	601.56	−2.57
48 to 84 months	2218.32	192.90	8	1543.18	461.84	546.62	−534.72
84 to 120 months	2972.91	258.51	5	1292.57	191.36	409.59	−691.62
>120 months	3979.03	346.00	1.75	605.50	157.32	273.31	−174.88
						Mean final balance	−270

* Cost of production per year of useful life = Cost of production/useful life (11.5 years); ** Animal unrecovered value = (Cost of production per year of useful life) × (Outstanding useful life at slaughter); *** Final balance by slaughter = (Cull value + Compensation received) − (Animal unrecovered value).

**Table 5 animals-10-02433-t005:** Loss from replacement on a farm with 25 head of cattle.

25-Head Cattle Farm						
Number of test-positive animals	0	1	2	3	4	5
Number of cows on farm	25	25	25	25	25	25
Number of cows to be slaughtered due to TB	0	1	2	3	4	5
Number of productive cows	25	24	23	22	21	20
Production considering fertility and mortality at birth *	19	18	17	16	16	15
Male calves	9	9	9	8	8	7
Female calves	10	9	8	8	8	8
Yearly replacement	3	3	3	3	3	3
Increased replacement due to TB		1	2	3	4	5
Number of male calves sold	9	9	9	8	8	7
Sale price of male calves	676.97	676.97	676.97	676.97	676.97	676.97
Number of female calves sold **	7	5	3	2	1	0
Sale price of female calves	458.1	458.1	458.1	458.1	458.1	458.1
Sales revenue	9299.4	8383.2	7467	6332	5873.9	4738.8
Annual reduction in sales revenue ***		−916.2	−1832.4	−2967.47	−3425.57	−4560.64
Reduction in sales revenue over two and a half years ****		−2290.5	−4581	−7418.68	−8563.93	−11,401.6

* (Number of productive cows × (9/11.5)) − ((Number of productive cows × 9/11.5) × 0.05); ** Number of female calves sold = Female calves − Yearly replacement − Increased replacement due to TB; *** Annual reduction in sales revenue (for n test-positive animals) = Sales revenue for n test-positive animals scenario − Sales revenue for 0 test-positive animals scenario; **** Reduction in sales revenue over two and a half years = Annual reduction in sales revenue × 2.5 years.

**Table 6 animals-10-02433-t006:** Loss from replacement on a farm with 60 head of cattle.

60-Head Cattle Farm							
Number of test-positive animals	0	1	2	3	4	5	10
Number of cows on farm	60	60	60	60	60	60	60
Number of cows to be slaughtered due to tuberculosis (TB)	0	1	2	3	4	5	10
Number of productive cows	60	59	58	57	56	55	50
Production considering fertility and mortality at birth *	45	44	43	42	42	41	37
Male calves	22	22	22	21	21	20	19
Female calves	23	22	21	21	21	21	18
Yearly replacement	8	8	8	8	8	8	8
Increased replacement due to TB		1	2	3	4	5	10
Number of male calves sold	22	22	22	21	21	20	19
Sale price of male calves	676.97	676.97	676.97	676.97	676.97	676.97	676.97
Number of female calves sold **	15	13	11	10	9	8	0
Sale price of female calves	458.1	458.1	458.1	458.1	458.1	458.1	458.1
Sales revenue	21,764.8	20,848.6	19,932.4	18,797.4	18,339.3	17,204.2	12,862.4
Annual reduction in sales revenue ***		−916.20	−1.832,40	−2967.47	−3425.57	−4560.64	−8902.41
Reduction in sales revenue over two and a half years ****		−2290.50	−4581.00	−7418.67	−8563.93	−11,401.60	−22,256.03

* (Number of productive cows × (9/11.5)) − ((Number of productive cows × 9/11.5) × 0.05); ** Number of female calves sold = Female calves − Yearly replacement − Increased replacement due to TB; *** Annual reduction in sales revenue (for n test-positive animals) = Sales revenue for n test-positive animals scenario − Sales revenue for 0 test-positive animals scenario; **** Reduction in sales revenue over two and a half years = Annual reduction in sales revenue × 2.5 years.

**Table 7 animals-10-02433-t007:** Loss from replacement on a farm with 100 head of cattle.

100-Head Cattle Farm											
Number of test-positive animals	0	1	2	3	4	5	10	15	16	17	18
Number of cows on farm	100	100	100	100	100	100	100	100	100	100	100
Number of cows to be slaughtered due to TB	0	1	2	3	4	5	10	15	16	17	18
Number of productive cows	100	99	98	97	96	95	90	85	84	83	82
Production considering fertility and mortality at birth *	74	74	73	72	71	71	67	63	62	62	61
Male calves	37	37	36	36	36	35	33	32	31	31	30
Female calves	37	37	37	36	35	36	34	31	31	31	31
Yearly replacement	13	13	13	13	13	13	13	13	13	13	13
Increased replacement due to TB		1	2	3	4	5	10	15	16	17	18
Number of male calves sold	37	37	36	36	36	35	33	32	31	31	30
Sale price of male calves	676.97	676.97	676.97	676.97	676.97	676.97	676.97	676.97	676.97	676.97	676.97
Number of female calves sold **	24	23	22	21	19	18	11	4	3	1	0
Sale price of female calves	458.1	458.1	458.1	458.1	458.1	458.1	458.1	458.1	458.1	458.1	458.1
Sales revenue	36,042.3	35,584.2	34,449.1	33,991.0	33,074.8	31,939.8	27,379.1	23,495.4	22,360.4	21,444.2	20,309.1
Annual reduction in sales revenue ***		−458.10	−1593.17	−2051.27	−2967.47	−4102.54	−8663.18	−12,546,85	−13,681.92	−14,598.12	−15,733.2
Reduction in sales revenue over two and a half years ****		−1145.25	−3982.93	−5128.17	−7418.68	−10,256.35	−21,657.95	−31,367.13	−34,204.8	−36,495.3	−39,332.9

* (Number of productive cows × (9/11.5)) − ((Number of productive cows × 9/11.5) × 0.05); ** Number of female calves sold = Female calves − Yearly replacement − Increased replacement due to TB; *** Annual reduction in sales revenue (for n test-positive animals) = Sales revenue for n test-positive animals scenario − Sales revenue for 0 test-positive animals scenario; **** Reduction in sales revenue over two and a half years = Annual reduction in sales revenue × 2.5 years.

**Table 8 animals-10-02433-t008:** Loss from replacement on a farm with 500 head of cattle.

500-Head Cattle Farm										
Number of test-positive animals	0	1	2	3	5	10	15	20	25	90
Number of cows on farm	500	500	500	500	500	500	500	500	500	500
Number of cows to be slaughtered due to TB	0	1	2	3	5	10	15	20	25	25
Number of productive cows	500	499	498	497	495	490	485	480	475	410
Production considering fertility and mortality at birth *	372	371	370	370	368	364	361	357	353	305
Male calves	186	185	185	185	184	182	180	178	177	152
Female calves	186	186	185	185	184	182	181	179	176	153
Yearly replacement	63	63	63	63	63	63	63	63	63	63
Increased replacement due to TB		1	2	3	5	10	15	20	25	90
Number of male calves sold	186	185	185	185	184	182	180	178	177	152
Sale price of male calves	676.97	676.97	676.97	676.97	676.97	676.97	676.97	676.97	676.97	676.97
Number of female calves sold **	123	122	120	119	116	109	103	96	88	0
Sale price of female calves	458.1	458.1	458.1	458.1	458.1	458.1	458.1	458.1	458.1	458.1
Sales revenue	182,262.7	181,127.7	180,211.5	179,753.4	177,702.1	173,141.4	169,038.9	164,478.3	160,136.5	102,899.4
Annual reduction in sales revenue ***		−1135.07	−2051.27	−2509.37	−4560.64	−9121.28	−13,223.82	−17,784.46	−22,126.23	−79,363.28
Reduction in sales revenue over two and a half years ****		−2837.67	−5128.17	−6273.42	−11,401.60	−22,803.20	−33,059.55	−44,461.15	−55,315.58	−198,408.20

* (Number of productive cows × (9/11.5)) − ((Number of productive cows × 9/11.5) × 0.05); ** Number of female calves sold = Female calves − Yearly replacement − Increased replacement due to TB; *** Annual reduction in sales revenue (for n test-positive animals) = Sales revenue for n test-positive animals scenario − Sales revenue for 0 test-positive animals scenario; **** Reduction in sales revenue over two and a half years = Annual reduction in sales revenue × 2.5 years.

**Table 9 animals-10-02433-t009:** Loss of profit on a farm with 25 head of cattle.

25-Head Cattle Farm						
Annual loss of profit:						
Number of test-positive animals	0	1	2	3	4	5
Mean final balance by slaughtered animal		−270.3	−540.6	−811	−1081.30	−1351.60
Reduction in sales revenue		−916.2	−1832.40	−2967.47	−3425.57	−4560.64
Incremental cost of rearing replacement calf		503.06	1006.12	1509.18	2012.24	2515.30
Annual loss of profit:		−1689.58	−3379.16	−5287.61	−6519.09	−8427.54
Overall loss of profit over two and a half years:						
Number of test-positive animals	0	1	2	3	4	5
Mean final balance by slaughtered animal		−270.3	−540.6	−811.0	−1081.3	−1351.6
Reduction in sales revenue		−2290.50	−4581.00	−7418.68	−8563.93	−11,401.60

Incremental cost of rearing replacement calf		1257.65	2515.30	3772.95	5030.60	6288.25
Overall loss of profit over two and a half years:		−3818.47	−7636.94	−12,002.58	−14,675.80	−19,041.45

**Table 10 animals-10-02433-t010:** Loss of profit on a farm with 60 head of cattle.

60-Head Cattle Farm							
Annual loss of profit:							
Number of test-positive animals	0	1	2	3	4	5	10
Mean final balance by slaughtered animal		−270.32	−540.64	−810.96	−1081.28	−1351.60	−2703.20
Reduction in sales revenue		−916.20	−1832.40	−2967.47	−3425.57	−4560.64	−8902.41
Incremental cost of rearing replacement calf		503.06	1006.12	1509.18	2012.24	2515.30	5030.60
Annual loss of profit:		−1689.58	−3379.16	−5287.61	−6519.09	−8427.54	−1689.58
Overall loss of profit over two and a half years:							
Number of test-positive animals	0	1	2	3	4	5	10
Mean final balance by slaughtered animal		−270.3	−540.6	−811.0	−1081.3	−1351.6	−2703.2
Reduction in sales revenue		−2290.50	−4581.00	−7418.67	−8563.93	−11,401.60	−22,256.03
Incremental cost of rearing replacement calf		1257.65	2515.30	3772.95	5030.60	6288.25	12,576.50
Overall loss of profit over two and a half years:		−3818.47	−7636.94	−12,002.58	−14,675.80	−19,041.45	−37,535.72

**Table 11 animals-10-02433-t011:** Loss of profit on a farm with 100 head of cattle.

100-Head Cattle Farm											
Annual loss of profit:											
Number of test-positive animals	0	1	2	3	4	5	10	15	16	17	18
Mean final balance by slaughtered animal		−270.32	−540.64	−810.96	−1081.28	−1351.60	−2703.20	−4054.79	−4325.10	−4595.40	−4865.80
Reduction in sales revenue		−458.10	−1593.17	−2051.27	−2967.47	−4102.54	−8663.18	−12,546.85	−13,681.92	−14,598.12	−15,733.2
Incremental cost of rearing replacement calf		503.06	1006.12	1509.18	2012.24	2515.30	5030.60	7545.90	8048.96	8552.02	9055.08
Annual loss of profit:		−1231.48	−3139.93	−4371.41	−6060.99	−7969.44	−16,396.98	−24,147.54	−26,055.99	−27,745.57	−29,654.02
Overall loss of profit over two and a half years:											
Number of test-positive animals	0	1	2	3	4	5	10	15	16	17	18
Mean final balance by slaughtered animal		−270.3	−540.6	−811.0	−1081.3	−1351.6	−2703.2	−4054.8	−4325.10	−4595.40	−4865.80
Reduction in sales revenue		−1145.25	−3982.93	−5128.17	−7418.68	−10,256.35	−21,657.95	−31,367.13	−34,204.8	−36,495.3	−39,332.9
Incremental cost of rearing replacement calf		1257.65	2515.30	3772.95	5030.60	6288.25	12,576.50	18,864.75	20,122.4	21.38	22,637.7
Overall loss of profit over two and a half years:		−2673.22	−7038.86	−9712.08	−13,530.55	−17,896.20	−36,937.65	−54,286.67	−58,652.31	−62,470.78	−66,836.43

**Table 12 animals-10-02433-t012:** Loss of profit on a farm with 500 head of cattle.

500-Head Cattle Farm										
Annual loss of profit:										
Number of test-positive animals	0	1	2	3	5	10	15	20	25	90
Mean final balance by slaughtered animal		−270.32	−540.64	−810.96	−1351.60	−2703.20	−4054.79	−5406.40	−6758.00	−24,328.80
Reduction in sales revenue		−1135.07	−2051.27	−2509.37	−4560.64	−9121.28	−13,223.82	−17,784.46	−22,126.23	−79,363.28
Incremental cost of rearing replacement calf		503.06	1006.12	1509.18	2515.30	5030.60	7545.90	10,061.20	12,576.50	45,275.40
Annual loss of profit:		−1908.45	−3598.03	−4829.51	−8427.54	−16,855.08	−24,824.51	−33,252.05	−41,460.72	−148,967.44
Overall loss of profit over two and a half years:										
Number of test-positive animals	0	1	2	3	5	10	15	20	25	90
Mean final balance by slaughtered animal		−270.3	−540.6	−811.0	−1351.6	−2703.2	−4054.8	−5406.40	−6758.00	−24,328.80
Reduction in sales revenue		−2837.67	−5128.17	−6273.42	−11,401.60	−22,803.20	−33,059.55	−44,461.15	−55,315.58	−198,408.20
Incremental cost of rearing replacement calf		1257.65	2515.30	3772.95	6288.25	12,576.50	18,864.75	25,153.00	31,441.25	113,188.50
Overall loss of profit over two and a half years:		−4365.64	−8184.11	−10,857.33	−19,041.45	−38,082.90	−55,979.09	−75,020.54	−93,514.81	−335,925.46

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
