# Peer review of "Quantifying the Economic Impact of Bovine Tuberculosis on Livestock Farms in South-Western Spain"

_animals, 2020, doi:10.3390/ani10122433_

Round 1
Reviewer 1 Report
Much improved and much clearer comapred to original submission
Author Response
Reviewer 1
Can be improved
|
Are the methods adequately described? |
( ) |
(x) |
( ) |
|
Are the results clearly presented? |
( ) |
(x) |
( ) |
Much improved and much clearer compared to original submission
- Are the methods adequately described? Can be improved.
In order to improve the Materials and Methods section we have restructured it. This section has been divided into two subheadings:
3.1. Materials and Methods for the estimation of cost of production.
3.2. Materials and Methods for the estimation of loss of profit.
In this section we have taken some paragraphs that were previously in the Background or in the Results section and that thanks to the reviewer's comments we see clearly that they should be in the Materials and Methods section. It can be seen in lines 264 to 391.
- Are the results clearly presented? Can be improved.
Also, section 4. Results, has been restructured as can be seen on lines 392 to 643.
In order to better clarify the results, we have split tables 5, 6 and 7 and 8 of the previous version into tables 5 to 12 in this new version. It can be seen in lines 442 to 494 and 518 to 596.
Thanks for your time and effort.

Reviewer 2 Report
a lot of material has been added that makes the paper more readable and understandable but it is still not easy to interpret
there are still spelling and grammatical errors and some sections are a little disjointed
some terminology needs to be clarified e.g line 144 ' covering' - mating?
the added background and conceptual framework section is not easy to understand and make sense of. may need to provide the detail in a more structured approach. much of the framework section is the method used and should be reflected in that section. some sub headings may be useful
some information in the methods section needs more discussion in that section e.g lines 342-345 loss of sales revenue could be higher - this is a discussion point; as is 346-351
similarly the results section has many discussion points. is there a way you can add formulas to the methods/results section? e.g useful life x purchase value = XX - (cull value + annual direct costs......)
some tables in the results are confusing and hard to interpret and it may be worth generating multiple tables that reflect the information more clearly
addition of graphs adds to the paper
Author Response
Reviewer 2
Comments and Suggestions for Authors
A lot of material has been added that makes the paper more readable and understandable but it is still not easy to interpret.
- There are still spelling, and grammatical errors and some sections are a little disjointed.
We have revised the spelling and grammar and have corrected the following:
-we have removed extra comas and spaces.
-former line 110: it said “fourth section” and should say “fifth section”.
-former line 236: “proposed” was duplicated.
-former line 478: it said “amounts” and it should be “amount”.
-former line 562: it said “is mitigated” and should be “are mitigated”.
-former line 318: it said “prices of 31 December” and should be “as of 31 December”
-former line 632: it said “south-west Spain” and should say “south-western Spain”
-“Inmobilization” and “inmobilisation” were used randomly and now we have unified them as “inmobilization”.
-former line 644: it said “foregone” and should be “forgone”.
- Some terminology needs to be clarified e.g line 144 ' covering' - mating?
We have changed the word “covering” for the word “mating” in current line 140.
- The added background and conceptual framework section are not easy to understand and make sense of. May need to provide the detail in a more structured approach. Much of the framework section is the method used and should be reflected in that section. Some subheadings may be useful.
In order to clarify section 2, Background and theoretical model, we have structured it into three sub-sections:
2.1. Business model.
2.2. Actions and consequences derived from detection of TB.
2.3. Theoretical model and conceptual framework.
You can see it in lines 114 to 262.
In the same way we have restructured the Materials and Methods section. This section has been divided into two subheadings:
3.1. Materials and Methods for the estimation of cost of production.
3.2. Materials and Methods for the estimation of loss of profit.
In this section we have taken some paragraphs that were previously in the Background or in the Results section and that thanks to the reviewer's comments we see clearly that they should be in the Materials and Methods section. It can be seen in lines 264 to 391.
Also, section 4. Results, has been restructured as can be seen on lines 392 to 643.
- Some information in the methods section needs more discussion in that section e.g lines 342-345 loss of sales revenue could be higher - this is a discussion point as is 346-351.
Former lines 342-345 are now in the Discussion Section in lines 765 and 766.
Former lines 346-351 have been located in the Discussion Section in lines727 to 750.
- Similarly, the results section has many discussion points. is there a way you can add formulas to the methods/results section? e.g useful life x purchase value = XX - (cull value + annual direct costs......).
We have eliminated the points of discussion that were in the results in the previous version. This has not meant any changes in the discussion section so as not to duplicate aspects already included.
We have added formulas in the results section to clarify the calculations made. It can be seen in lines 422 to 424, 460 to 462, 471 to 473, 483 to 485 and 491 to 593.
- Some tables in the results are confusing and hard to interpret and it may be worth generating multiple tables that reflect the information more clearly.
In order to better clarify the results, we have split tables 5, 6 and 7 and 8 of the previous version into tables 5 to 12 in this new version. It can be seen in lines 442 to 494 and 518 to 596.
Thanks for your time and effort.

Reviewer 3 Report
In my opinion, the authors have done a good job in addressing the reviewers comments. The paper is much improved and I now consider it to be suitable for publication in Animals.
Author Response
Reviewer 3
Comments and Suggestions for Authors
In my opinion, the authors have done a good job in addressing the reviewer’s comments. The paper is much improved, and I now consider it to be suitable for publication in Animals.
We are very grateful for the comments you made in the first round to improve the work.
We also thank you for your kind words in this second review.
Thanks for your time and effort.

This manuscript is a resubmission of an earlier submission. The following is a list of the peer review reports and author responses from that submission.
Round 1
Reviewer 1 Report
This paper is very interesting and covers an issue of great concern to the Spanish economy.
However, I have a number of concerns about the paper
It is not clear what the objectives of the paper are intended to be. It is clearly to do with compensation to farmers whose animals have to be slaughtered consequent on TB infections. Having said that, it is not clear if the paper is suggesting a new framework for compensation or commenting on the weaknesses of the current system.
The paper outlines three types of losses
- Losses generated by the slaughter of infected cattle
- Lower revenues due to the non-sale of animals to replace the slaughtered cattle
- Costs incurred from immobilising the infected animals from the rest of the herd
Regarding 1, it seems to me that the losses here would be the cost of the calf (less cull value) plus the costs incurred in looking after the calf over a period of years. I am not sure compensation should include the loss of future profits since that is uncertain and is a risk and consequence of being in business. I cannot see any other industry getting compensation for this.
Regarding 2, I have a problem here. If farmers are compensated for the loss of the slaughtered animal then they have a choice of replacing the animal by another animal due for sale or purchasing a new animal. Having been compensated for the loss of the slaughtered animal, I fail to see why there is a further loss here. Maybe I am mis-understanding the process – in which case it needs to be clarified
Regarding item 3, I understand that it is difficult to get data but I cannot see that the paper is complete without even an estimate of the magnitude of the costs involved.
The paper seems to lack a theoretical framework for identifying the loss when an asset is lost. One possibility here is that of Deprival Value (see WT Baxter “Accounting Values and Inflation" 1975)
The paper has a lot of data and complex calculations which are difficult to follow even for someone with a strong financial background. Before re-submitting, I suggest the authors get the paper read by some informed colleagues to establish the ease of understanding
Author Response
Reviewer 1
This paper is very interesting and covers an issue of great concern to the Spanish economy.
We are very grateful for the kind comments you have made on the paper which will certainly help to improve it.
The changes shown in blue in the document are the result of the reviewers' assessments. We will now reply point by point to each of the comments made, indicating the lines of the paper in which this reply is reflected.
- It is not clear what the objectives of the paper are intended to be. It is clearly to do with compensation to farmers whose animals have to be slaughtered consequent on TB infections. Having said that, it is not clear if the paper is suggesting a new framework for compensation or commenting on the weaknesses of the current system.
We have tried to set out more clearly the objective set out in the paper. This objective is to carry out an estimate of the economic impact that tuberculosis (TB) disease at farm-level identified as the loss of profit on cattle farms. The contribution consists of providing a methodology for estimating this loss of profit on meat-producing cattle farms that apply the closed-herd policy model, in which replacement takes place with animals born on the farm itself.
In this sense, knowledge of the economic impact of the disease is relevant in the existing debate on the co-sharing of costs between government and farmers.
Also, in the Abstract we have tried to clarify better the objective pursued.
We have explained this in lines 81 to 106 of the Introduction.
In addition, in the section of Conclusions in lines 657 to 661, the authors explain the importance of quantifying the impact of TB on beef farms as an useful tool to help farmers make decisions on the advisability of investing in, and using, the measures of biosafety and biosecurity available to curb the disease on their farms. It would also be a useful tool for public administrations to help them calculate the compensation to be awarded to farmers to mitigate their losses.
The paper outlines three types of losses
- Losses generated by the slaughter of infected cattle
- Lower revenues due to the non-sale of animals to replace the slaughtered cattle
- Costs incurred from immobilising the infected animals from the rest of the herd
- Regarding 1, it seems to me that the losses here would be the cost of the calf (less cull value) plus the costs incurred in looking after the calf over a period of years. I am not sure compensation should include the loss of future profits since that is uncertain and is a risk and consequence of being in business. I cannot see any other industry getting compensation for this.
The paper posits that loss of profit is a broader concept than the loss from the slaughter of the animal. This is a matter of debate between livestock farmers and the administration, with some Spanish autonomous communities already recognizing the concept of loss of profit. To explain this better, please see lines 560 to 573 of the Discussion.
The loss from the slaughter referred to in the paper does not provide for compensation for future income. The compensation that farmers receive is regulated by RD 389/2011 of 18 March which establishes the compensation scales for the slaughter of animals within the framework of national programmes for the control or eradication of bovine tuberculosis. The amounts have been calculated taking into account market developments and in line with the provisions of European regulations.
Throughout the paper an attempt has been made to explain what loss by slaughter consists of. They appear in section 2 Background and Conceptual Framework on lines 243 to 250. The results section also explains how this loss has been estimated, specifically in table 4. Lines 387 to 400 explain this component.
- Regarding 2, I have a problem here. If farmers are compensated for the loss of the slaughtered animal, then they have a choice of replacing the animal by another animal due for sale or purchasing a new animal. Having been compensated for the loss of the slaughtered animal, I fail to see why there is a further loss here. Maybe I am mis-understanding the process – in which case it needs to be clarified.
The previous version did not explain sufficiently that the study is focused on farms that apply a closed-herd policy. This means replacements should be made with calves born within the herd, with these no longer being available for sale.
We apologize for this.
In order to clarify this point, we have created a new section, number 2. Background and Conceptual Framework (in particular see lines 127-153).
- Regarding item 3, I understand that it is difficult to get data but I cannot see that the paper is complete without even an estimate of the magnitude of the costs involved.
With regard to the immobilisation costs, we now have explained in the paper why this could not be addressed at this time (lines 346 to 351).
However, although we have not estimated it, we believe that it should be considered as a component of the loss of profit generated as it is a real and quantifiable cost.
Besides, lines 597 to 620 in the Discussion section give further insights as to the difficulties to estimate these costs considering the methodology applied. We mention other authors contribution to this topic using a different approach to ours. Finally, this has been recognized as a limitation of our analysis.
- The paper seems to lack a theoretical framework for identifying the loss when an asset is lost. One possibility here is that of Deprival Value (see WT Baxter “Accounting Values and Inflation" 1975)
Section 2. Background and Conceptual Framework has been incorporated in an attempt to solve the deficiency pointed out by the reviewer. Lines 167 to 273 contain the theoretical economic and accounting framework applied in the paper. Figures 1 and 2 have been added to illustrate the model for estimating costs of production and loss of profit, respectively.
- The paper has a lot of data and complex calculations which are difficult to follow even for someone with a strong financial background. Before re-submitting, I suggest the authors get the paper read by some informed colleagues to establish the ease of understanding
We admit that results needed further explanation and apologize for this.
All estimates have been explained again in more detail. The sections Materials and Methods and Results have been strongly restructured and expanded. This can be seen between lines 275 and 455.
We have also followed the reviewer's advice and asked other colleagues to read the paper to ensure their understanding. We hope that we have succeeded.
Thanks for your time and effort.

Reviewer 2 Report
some grammatical correction required throughout the paper
abstract does not clearly outline the economic impact in Spain - no direct results mentioned
no mention of pathophysiology of TB, why animal needs to be slaughtered, lesions for decrease in meat value, why TB hard to eradicate, average age when TB affects animal if chronic infection
tables require units to be added - live weight is deemed average LW (kg)??
many of the tables are confusing and hard to interpret
heading on table 5 missing
categories of production classes need to be clearly defined - 2 categories of calves should be split into un-weaned and weaned calves
much of what is discussed in results is the method used or a discussion on why that method was used; results not clearly defined
no mention of TB testing costs to farmer or ongoing costs required
why are animals due for replacement unable to be sold
no
Author Response
Reviewer 2
We are very grateful for the kind comments you have made on the paper which will certainly help to improve it. The changes shown in blue in the document are the result of the reviewers' assessments. We will now reply point by point to each of the comments made, indicating the lines of the paper in which this reply is reflected.
- Some grammatical correction required throughout the paper.
Once all the proposed changes have been made, a grammar check has been carried out by a native expert.
- Abstract does not clearly outline the economic impact in Spain - no direct results mentioned.
The abstract has been modified. You can see it in 32 to 40.
- No mention of pathophysiology of TB, why animal needs to be slaughtered, lesions for decrease in meat value, why TB hard to eradicate, average age when TB affects animal if chronic infection.
Noted and amended. We now write in lines 53 to 64.
- Tables require units to be added - live weight is deemed average LW (kg)??
Noted and amended. The tables include the measurement units.
- Many of the tables are confusing and hard to interpret.
The sections Materials and Methods and Results have been strongly restructured and expanded, and most of the tables have been modified for better understanding . This can be seen between lines 275 and 486.
- Heading on table 5 missing.
Noted and amended. Please see line 445: “60-head cattle farm”
- Categories of production classes need to be clearly defined – 2 categories of calves should be split into un-weaned and weaned calves.
Noted and amended. This is shown in table 1 and explained in lines 295 to 306.
- Much of what is discussed in results is the method used or a discussion on why that method was used; results not clearly defined.
We admit that results needed further explanation and apologize for this. All estimates have been explained again in more detail.
Method, Results and Discussion section have been extensively modified and properly separated. We hope that we have improved their understanding. It can be seen on lines 275 to 637.
- No mention of TB testing costs to farmer or ongoing costs required.
In Spain, the costs of the compulsory full-herd TB-testing are covered by the administration through the EU-cofounded National TB Control Scheme (Programa Nacional De Erradicación De Tuberculosis Bovina presentado por España para El Año 2019. Ministerio de Agricultura Pesca y Alimentación (MAPA) - Dirección General de Sanidad de la Producción, Madrid, Spain (last access 09 November 2020). https://www.mapa.gob.es/es/ganaderia/temas/sanidad-animal-higiene-ganadera/programatb2019verdefinitiva_tcm30-500265.pdf), it is the reference number 31.
This is reflected in lines 165 to 166 of the paper.
- Why are animals due for replacement unable to be sold?
The previous version did not explain sufficiently that the study is focused on farms that apply a closed-herd policy. This means replacements should be made with calves born within the herd, with these no longer being available for sale.
We apologize for this.
In order to clarify this point, we have created a new section, number 2. Background and Conceptual Framework (in particular see lines 127-153).
Thanks for your time and effort.

Reviewer 3 Report
Review of Manuscript ID: animals-988221 Quantifying the economic impact of bovine tuberculosis
on livestock farms in south-eastern Spain
This paper evaluates the economic impact at farm level of outbreaks of bovine TB on farms in Spain,
using data collated from a range of sources. In my opinion it is a useful and valuable contribution to
the literature and addresses an aspect of bovine TB which has not received sufficient attention.
However, there are certain methodological and other problems with the paper which need to be
addressed, and once they are addressed satisfactorily then I think this paper would be suitable for
publication in Animals.
1. The authors should put their work in a wider international context. While the work is of
course focussed on the specifics of Spain, it is likely to be of interest to a much wider
international audience, not because of the actual numerical results which are specific to
Spain, but because of the methodology and analytical approach used. It is good to see
Bicknell et al cited. The authors should add a paragraph to the introduction reviewing briefly
other work in this general area, including Bicknell et al; the recent UK Defra report on the
economic cost of a TB breakdown carried out by SAC (available here
http://randd.defra.gov.uk/Default.aspx?Menu=Menu&Module=More&Location=None&Proj
ectID=19957&FromSearch=Y&Publisher=1&SearchText=se3139&SortString=ProjectCode&So
rtOrder=Asc&Paging=10&s=09#Description) , and Barnes et al 2015 Preventive Veterinary
Medicine 122: 42-52. They should make the point that there are a range of methodologies
for looking at the cost of a TB breakdown and then also for looking at how these costs relate
to incentives and the challenge of deciding the balance of cost-sharing between the state
and the private sector, and that this paper adds to that body of knowledge by providing a
methodology and worked example from a TB-endemic region of Europe.
2. Line 20: losses due to non-sale of animals to replace slaughtered cattle: This wording is
confusing. If the animals are sold, they would not be replacing slaughtered cattle. Do the
authors mean “the income forgone from retaining within the herd cattle which would
otherwise have been sold, as they are needed to replace the slaughtered animals”?
3. Line 22 & 23: “demonstrate that” and “by no means sufficient”: I suggest changing to
“evaluating whether” and “is sufficient”. Asserting that it is by no means sufficient is a value
judgement implying that the state ought to bear a greater shar of the costs, which is not
immediately obvious.
4. Lines 32-33: same comment as above re phrasing of non-sale of replacements.
5. Line 35: “not sufficient” – change to “not sufficient to fully offset all costs”. Stating that
current compensation is not sufficient is a value judgement.
6. Lines 56-57: this statement needs to be balanced by a statement such as “the optimum
balance of cost-sharing between the state and the private sector remains a subject of
debate, as in other countries”, otherwise it is just a politicised value judgement which is not
suitable for a scientific paper.
7. Line 3 and line 75: The title of the paper references south eastern Spain, yet this refers to
Western Spain? I think it is very important that the authors correctly describe the study area
in the title and the text.
8. Line 88: Using the mean amount of subsidies for all farms will lead to overestimating its
contribution for small farms and underestimating it for large farms. The authors should
relate the subsidy received to the farm size. If this is not possible, they should provide a
range of subsidies (e.g. mean and upper and lower quartiles or similar) and address this
point in the discussion as a weakness in the analysis.
9. Lines 212-215: the authors do not explain why they chose herd sizes of 60, 100 and 500. If
60 is the average (and they should clarify – is this the mean or median?) why not select
another category below average? It seems strange to only study farms of average or higher
size. For context, they should provide data on the range of herd sizes in the study area –
mean, and upper and lower 25 th /75 th quartiles. Also, this should be in the materials and
methods, not results. The authors should add another category for farms below average size
and repeat the analysis for that herd size.
10. Line 122-137: This section mixes up the materials and methods with the results. The authors
should describe what they did in the materials and methods, and the results in the results
section; this is poor scientific writing.
11. Line 140-148: again, confusing M&M and results, poor drafting.
12. Table 4: the headings are confusing and do not match well with the descriptions of the
columns at the base of the table. Is “cull value” the same as “meat revenue”? Why use
different terms? Final balance is a better term than loss due to slaughter, as a positive loss
may still be seen by the reader as a loss rather than a gain.
13. Line 208: sale of test positive cattle? Surely test positive cattle must be slaughtered? Please
explain.
14. Line 222: The table shows analysis for test-positive animals from 0 to 25, but this may lead
the reader to think that high numbers of reactors are common. In most TB endemic areas,
most outbreaks involve just a few test-positive cattle. To put this in context, the authors
should provide a table showing the number of TB infected cattle per infected farm for the
region or else for Spain for the study period. For example, perhaps it may be that 25% of
outbreaks have 1-3 test-positive cattle, 50% have 4-8 and 25% have 9 or more. This would
put the scale of the losses implied by the table in a clearer context. Not that many outbreaks
would have 20 or 25 infected animals, in most cases.
15. Line 281: West or East? Is the title incorrect?
16. Line 288: define what an uncertified feedlot is and how they operate.
17. Line 297: Is Iberian a correct nomenclature for TB? I do not think this is an actual
bacteriological name for a strain of M bovis?
18. Line 306: increasing herd size is consistently shown to be a risk factor for TB, so please
explain how this will help reduce TB? Or maybe the idea is that while TB risk would increase,
the resilience of the farm in economic terms would be higher? If so, please state that.
19. Lines 338-339: this is a political value judgement, not a scientific conclusion, and should be
removed or reworded as per previous suggestions.
Author Response
Reviewer 3
This paper evaluates the economic impact at farm level of outbreaks of bovine TB on farms in Spain, using data collated from a range of sources. In my opinion it is a useful and valuable contribution to the literature and addresses an aspect of bovine TB which has not received sufficient attention.
However, there are certain methodological and other problems with the paper which need to beaddressed, and once they are addressed satisfactorily then I think this paper would be suitable for publication in Animals.
We are very grateful for the kind comments you have made on the paper which will certainly help to improve it. The changes shown in blue in the document are the result of the reviewers' assessments. We will now reply point by point to each of the comments made, indicating the lines of the paper in which this reply is reflected.
- The authors should put their work in a wider international context. While the work is of course focused on the specifics of Spain, it is likely to be of interest to a much wider international audience, not because of the actual numerical results which are specific to Spain, but because of the methodology and analytical approach used. It is good to see Bicknell et al cited. The authors should add a paragraph to the introduction reviewing briefly other work in this general area, including Bicknell et al; the recent UK Defra report on the
economic cost of a TB breakdown carried out by SAC (available here
http://randd.defra.gov.uk/Default.aspx?Menu=Menu&Module=More&Location=None&ProjectID=19957&FromSearch=Y&Publisher=1&SearchText=se3139&SortString=ProjectCode&SortOrder=Asc&Paging=10&s=09#Description) , and Barnes et al 2015 PreventiveVeterinaryMedicine 122: 42-52. They should make the point that there are a range of methodologies for looking at the cost of a TB breakdown and then also for looking at how these costs relate to incentives and the challenge of deciding the balance of cost-sharing between the state and the private sector, and that this paper adds to that body of knowledge by providing a methodology and worked example from a TB-endemic region of Europe.
We have reinforced the contribution of the paper and provided previous literature in the field from an international perspective. This is shown in the introduction (lines 81 to 106). In the discussion section, our results have been commented in relation to previous research (lines 542 to 629).
- Line 20: losses due to non-sale of animals to replace slaughtered cattle: This wording is If the animals are sold, they would not be replacing slaughtered cattle. Do theauthors mean “the income forgone from retaining within the herd cattle which wouldotherwise have been sold, as they are needed to replace the slaughtered animals”?
- Lines 32-33: same comment as above rephrasing of non-sale of replacements.
We will response jointly to points 2 and 4.
Noted and amended in lines at simple summary (22-23), abstract (lines 33-34) and at the introduction (lines 100-102).
This component of the loss is better explained now in the Results (lines 462-475).
- Line 22 & 23: “demonstrate that” and “by no means sufficient”: I suggest changing to “evaluating whether” and “is sufficient”. Asserting that it is by no means sufficient is a value judgement implying that the state ought to bear a greater share of the costs, which is not immediately obvious.
- Line 35: “not sufficient” – change to “not sufficient to fully offset all costs”. Stating that current compensation is not sufficient is a value judgement.
- Lines 56-57: this statement needs to be balanced by a statement such as “the optimum balance of cost-sharing between the state and the private sector remains a subject of debate, as in other countries”, otherwise it is just a politicised value judgement which is not suitable for a scientific paper.
We will address jointly the response to points 3, 5 and 6.
Noted and amended, in the simple summary we have added lines 26 to 27, in abstract in lines 37 to 40.
In the introduction, this idea is conveyed in lines 78-79, and a reflection on this issue is also provided in the discussion, lines 560-573. The contribution to this debate is referred to in the conclusion, lines 657-661.
- Line 3 and line 75: The title of the paper references south eastern Spain, yet this refers to Western Spain? I think it is very important that the authors correctly describe the study area in the title and the text.
- 15. Line 281: West or East? Is the title incorrect?
We apologize for this mistake referred to in points 7 and 15. The title has been corrected and now refers to south-western Spain.
The region of study is described in the Method and Materials section (lines 276-278).
- Line 88: Using the mean amount of subsidies for all farms will lead to overestimating itscontribution for small farms and underestimating it for large farms. The authors should relate the subsidy received to the farm size. If this is not possible, they should provide a range of subsidies (e.g. mean and upper and lower quartiles or similar) and address this point in the discussion as a weakness in the analysis.
In the text, lines 313-325 explain that.
We are aware that this data shows a great variability depending on the type and size of farms, being therefore a limitation of the research. However, as we do not have this information by farm, but by head of livestock, we had to use the average of these values between 2013 and 2017, which is 159.76 euros/head. The available data shows its minimum value in the year 2015 in the region of Castilla-León with an amount of 79.46 euros/head, and the maximum in Andalusia in the year 2014 with an amount of 216.42 euros/head.
Following your advice, we have referred to this point as a limitation of the analysis within the discussion (lines 630 a 633).
- Lines 212-215: the authors do not explain why they chose herd sizes of 60, 100 and 500. If 60 is the average (and they should clarify – is this the mean or median?) why not select another category below average? It seems strange to only study farms of average or higher For context, they should provide data on the range of herd sizes in the study area –mean, and upper and lower 25 th /75 th quartiles. Also, this should be in the materials and methods, not results. The authors should add another category for farms below average size and repeat the analysis for that herd size.
Noted and amended. We have added to the study the case of a farm of 25 heads. Besides, we now explain how the selection of the sizes studied was carried out (lines 407 to 415).
- Line 122-137: This section mixes up the materials and methods with the results. The authorsshould describe what they did in the materials and methods, and the results in the resultssection; this is poor scientific writing.
- Line 140-148: again, confusing M& M and results, poor drafting.
We reply jointly to points 10 and 11.
We apologize for this. In the new version sections 3 (Materials and Methods) and 4 (Results) have been heavily restructured and expanded.
- Table 4: the headings are confusing and do not match well with the descriptions of the columns at the base of the table. Is “cull value” the same as “meat revenue”? Why use different terms? Final balance is a better term than loss due to slaughter, as a positive loss may still be seen by the reader as a loss rather than a gain.
We have eliminated different terms for the same concepts throughout the text. In particular “meat revenue” does no longer appear and is referred to as “cull value”.
Table 4 has been modified and its content has been extended for better understanding (lines 387 to 400).
We have used the term “final balance” instead of “loss” when referring to the economic effects due to the slaughter of the infected animal throughout the text (lines 21, 99, 125, 239 figure 1, 387, 394, 457, etc.)
- Line 208: sale of test positive cattle? Surely test positive cattle must be slaughtered? Please explain.
This was poorly explained. Lines 154-164 now explain this better. Positive cattle must be slaughtered within 30 days. The rest of the herd is subject to restrictions on movement, only being allowed to leave the farm for sale to an uncertified feedlot (Royal Decree 727/2011).
- Line 222: The table shows analysis for test-positive animals from 0 to 25, but this may lead the reader to think that high numbers of reactors are common. In most TB endemic areas, most outbreaks involve just a few test-positive cattle. To put this in context, the authors should provide a table showing the number of TB infected cattle per infected farm for the region or else for Spain for the study period. For example, perhaps it may be that 25% of outbreaks have 1-3 test-positive cattle, 50% have 4-8 and 25% have 9 or more. This would
put the scale of the losses implied by the table in a clearer context. Not that many outbreaks would have 20 or 25 infected animals, in most cases.
Noted. The reviewer is right in indicating that most outbreaks have few positive cattle. However, this is the case mainly if there is no culture-confirmation. When there is culture-confirmation, the more sensitive gamma interferon test is used in addition to the skintest, and when using this additional test, it is no longer exceptional to find larger numbers of TB-positive cattle. This was now added to lines 417 to 422 of the discussion.
This was also referred to in the introduction (lines 65 to 67).
- Line 288: define what an uncertified feedlot is and how they operate.
Noted and amended. This has been changed in the Introduction (lines 92 to 95), in the Background and conceptual framework (lines 154 to 164), the Results (lines 341 to 348) and the Discussion (590 a 596).
- Line 297: Is Iberian a correct nomenclature for TB? I do not think this is an actual
bacteriological name for a strain of M bovis?
Agreed and modified. It was incorrect wording. In lines 522 to 524, instead of “actions aimed at the primary maintenance hosts of the Iberian Mycobacterium TB complex, namely, the wild ungulates, including wild boar, red deer and fallow deer [31]” we now write “actions aimed at the primary maintenance hosts of the MTC in the Iberian Peninsula, namely, the wild ungulates, including wild boar, red deer and fallow deer [31]”
- Line 306: increasing herd size is consistently shown to be a risk factor for TB, so please
explain how this will help reduce TB? Or maybe the idea is that while TB risk would increase, the resilience of the farm in economic terms would be higher? If so, please state that.
We agree with the idea that while TB risk would increase,
the resilience of the farm in economic terms would be higher. This fact is pointed out in the Results (lines 493 to 497) and the Discussion (lines 621 to 625) and in the Conclusion (lines 654 to 656). - Lines 338-339: this is a political value judgement, not a scientific conclusion, and should be removed or reworded as per previous suggestions.
This point has been addressed as a response to points 3, 5 and 6.
Noted and amended, in the simple summary we have added lines 26 to 27, in abstract in lines 37 to 40.
In the introduction, this idea is conveyed in lines 78-79, and a reflection on this issue is also provided in the discussion, lines 560-573. The contribution to this debate is referred to in the conclusion, lines 657-661.
Thanks for your time and effort.
